# Tight Rates in Supervised Outlier Transfer Learning

**Mohammadreza M. Kalan**
Statistics, Columbia University
mm6244@columbia.edu

**Samory Kpotufe**
Statistics, Columbia University
samory@columbia.edu

## Abstract

A critical barrier to learning an accurate decision rule for outlier detection is the scarcity of outlier data. As such, practitioners often turn to the use of similar but imperfect outlier data from which they might *transfer* information to the target outlier detection task. Despite the recent empirical success of transfer learning approaches in outlier detection, a fundamental understanding of when and how knowledge can be transferred from a source to a target outlier detection task remains elusive. In this work, we adopt the traditional framework of Neyman-Pearson classification—which formalizes *supervised outlier detection*—with the added assumption that one has access to some related but imperfect outlier data. Our main results are as follows:

- We first determine the information-theoretic limits of the problem under a measure of discrepancy that extends some existing notions from traditional balanced classification; interestingly, unlike in balanced classification, seemingly very dissimilar sources can provide much information about a target, thus resulting in fast transfer.
- We then show that, in principle, these information theoretic limits are achievable by *adaptive* procedures, i.e., procedures with no a priori information on the discrepancy between source and target outlier distributions.

## 1 Introduction

A primary objective in many data science applications is to learn a decision rule that separates a common class with abundant data from a *rare* class with limited or no data. This is a traditional problem which often appears under the umbrella term of *outlier detection* or *rare class classification*, and has seen a resurgence of interest in modern applications such as malware detection in cybersecurity and IoT (Jose et al., 2018; Kumar & Lim, 2019), fraud detection in credit card transactions (Malini & Pushpa, 2017), disease diagnosis (Bourzac, 2014; Zheng et al., 2011), among others. A main goal in these applications—which distinguishes it from traditional classification where performance is asssessed on *average* over all classes—is to achieve low classification error on the rare class, while at the same time maintaining low error w.r.t. the common class. Such a constrained objective is commonly referred to as Neyman-Pearson classification. Formally, letting $\mu_0, \mu_1$ denote the common and rare class distributions, Neyman-Pearson classification takes the form:

> Minimize $\mu_1$-error over classifiers $h$ in some hypothesis space $\mathcal{H}$
>
> subject to keeping the $\mu_0$-error of such an $h$ under a threshold $\alpha$.

In this work, we focus on the common supervised setting where practitioners have access to not only training data from the common class, but also some (limited amount of) data from the rare class or, pertinently, *from a related distribution they hope has information on the rare class*. Henceforth, for simplicity of notation, we denote the *target* rare class distribution by $\mu_{1,T}$ and the related but imperfect rare class distribution by $\mu_{1,S}$, where "S" stands for *source*. As an example, such related rare-class data may be from a different infected device in IoT applications, or from similar cancer types in medical applications, or from laboratory simulations of a rare class. This is thus a *transfer learning* problem, however for *supervised outlier detection* rather than for traditional classification as is usual in the literature.

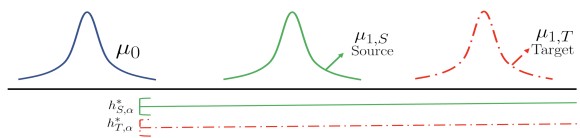

Figure 1: $\mu_0 = \mathcal{N}(a_0, \sigma^2)$ is the common distribution, and $\mu_{1,S}, \mu_{1,T} = \mathcal{N}(a_{1,S}, \sigma^2), \mathcal{N}(a_{1,T}, \sigma^2)$ are the source and target distributions. The universally optimal decision rules for the source and target are identical, i.e., $h_{S,\alpha}^* = h_{T,\alpha}^*$, in fact for any value of $\alpha \in [0, 1]$ (see Section 3).

While this *outlier transfer* problem is quite common in applications due to the scarcity of rare class data (Su et al., 2023; Wen & Keyes, 2019; Aburakhia et al., 2020), the problem has so far received little rigorous attention. Our main aim is therefore to further theoretical understanding of this timely problem, and in particular to gain much needed insight on the extent to which related but imperfect rare class data may improve performance for the *target* Neyman-Pearson classification problem. Such achievable transfer performance of course must depend on *how far* the related rare class distribution $\mu_{1,S}$ is from the target $\mu_{1,T}$—somehow properly formalized—and or whether the related rare-class distribution $\mu_{1,S}$ induces similar optimal decision rules as for the target rare class distribution $\mu_{1,T}$.

We first argue that, unlike in traditional classification, seemingly very different source and target distributions $\mu_{1,S}, \mu_{1,T}$ may induce the same exact (universally) optimal decision rules in Neyman-Pearson classification; this is obtained as a consequence of a simple extension of the classical Neyman-Pearson Lemma (Lehmann & Lehmann, 1986) to the case of transfer (see Proposition 3.6). This is illustrated in Fig 1 and explained in detail in Section 3. As a consequence, unlike in usual classification, we can approximate the universally best classifier $h_{T,\alpha}^*$ under the target arbitrarily well asymptotically, i.e., with sufficiently large data from a seemingly unrelated source.

However, the story turns out more nuanced in finite sample regimes, i.e, as we consider the rate of such approximation (in terms of relevant sample sizes), even when $\mu_{1,S}$ and $\mu_{1,T}$ admits the same optimal classifiers. That is, two different sources $\mu_{1,S}$ and $\mu_{1,S}'$ may yield considerably different transfer rates in finite-sample regimes even if both of them share the same optimal classifier as the target $\mu_{1,T}$: this is because a given source may yield more data near the common decision boundary $h_{T,\alpha}^*$ than another source, thus revealing $h_{T,\alpha}^*$ at a faster rate. In particular, we show in our first main result of Theorem 4.5—a minimax lower-bound—that the rate of convergence of *any outlier-transfer approach* is in fact controlled by a relatively simple notion of *outlier-transfer-exponent* (adapted from transfer results in traditional classification) which essentially measures how well a source may reveal the unknown decision boundary. Theorem 4.5 is in fact rather general: the minimax lower-bound holds for *any hypothesis space* $\mathcal{H}$ of finite VC dimension (at least 3), and any number of sample sizes from $\mu_0, \mu_{1,S}, \mu_{1,T}$, including the case of no sample from the rare target class $\mu_{1,T}$. Moreover, the result holds generally $h_{S,\alpha}^*$ and $h_{T,\alpha}^*$ are the same or not.

We finally turn our attention to whether such rates may be achieved *adaptively*, i.e., from samples alone without prior knowledge of the discrepancy between $\mu_{1,S}$ and $\mu_{1,T}$ as captured by both the transfer-exponent and the *amount* of difference between optimal classifiers $h_{S,\alpha}^*$ and $h_{T,\alpha}^*$. We show in Theorem 4.6 that this is indeed the case: the minimax lower-bounds of Theorem 4.5 can be matched up to logarithmic factors by some careful adaptation approach that essentially *fine-tunes* classifiers learned using source data with whatever target data is available. This is described in Section 4.8.

## 1.1 RELATED WORK

Outlier detection and transfer learning have mostly been studied separately, despite the clear need for transfer learning in applications of outlier detection where the rare class of data is by definition, always scarce.

As such, transfer learning works have mostly focused on traditional classification and regression starting from seminal works of Mansour et al. (2009); David et al. (2010); Ben-David et al. (2010; 2006), to more recent works of Hanneke & Kpotufe (2019; 2022); Kalan et al. (2022); Mousavi Kalan et al. (2020); Lei et al. (2021). The works of Hanneke & Kpotufe (2019; 2022) are most closely related as our notion of outlier-transfer-exponent may be viewed as an extension

of their notion of transfer-exponent; however, besides for the fact that both notions capture discrepancies around decision boundaries, transfer in outlier detection is fundamentally different from the case of traditional classification studied in the above works: for instance, as stated earlier, distributions that are significantly different in traditional classification can be quite close in outlier transfer as revealed in this work.

Theoretical works on outlier detection on the other hand have mostly focused on unsupervised and supervised settings, but without considering the more practical transfer setting. Unsupervised outlier-detection assumes that only data from the common class $\mu_0$ is available; theoretical works include studies of density-level set estimation (Steinwart et al., 2005; Polonik, 1995; Ben-David & Lindenbaum, 1997; Tsybakov, 1997) where *outliers* are viewed as data in low density regions, or in works on so-called *one-class classification* that aim to learn a contour of the common class $\mu_0$ (Schölkopf et al., 2001). Supervised outlier-detection has commonly been formalized via Neyman-Pearson classification, where some data from both the common and rare classes are used to optimize and constrain empirical errors. Early works include (Cannon et al., 2002; Scott & Nowak, 2005; Blanchard et al., 2010; Rigollet & Tong, 2011) which establish convergence rates in various distributional and model selection settings, but all exclude the question of transfer.

Transfer learning for outlier detection has in fact received much recent attention in the methodological literature (Xiao et al., 2015; Andrews et al., 2016; Idé et al., 2017; Chalapathy et al., 2018; Yang et al., 2020) where various approaches have been proposed that aim to leverage shared structural aspects of source and target rare class data.

On the theoretical front however, much less is understood about outlier transfer. The recent work of Scott (2019) initiates theoretical understanding of the problem: they are first to show that, in some situations where both source and target share the same optimal classifiers, various procedures can guarantee consistency (i.e., taking sample size to infinity) even as source and target $\mu_{1,S}, \mu_{1,T}$ appear different. Our Proposition 3.6 shows that in fact optimal classifiers may be shared in even more general situations, similarly implying consistency for seemingly very different source and target rare-class distributions. Our main results of Theorems 4.5 and 4.6 reach further in deriving the first insights into the finite-sample regimes of outlier-transfer, by establishing information-theoretic limits of the problem, along with notions of discrepancies between source and target that tightly capture such limits.

## 2 SETUP

We first formalize the Neyman-Pearson classification framework, followed by its extension to the transfer case.

### 2.1 NEYMAN-PEARSON CLASSIFICATION

Let $\mu_0$ and $\mu_1$ denote probability distributions on some measurable space $(\mathcal{X}, \Sigma)$. Furthermore, suppose that $\mathcal{H}$ is a hypothesis class consisting of measurable 0-1 functions on the domain $\mathcal{X}$, where we view $h(x) = 0$ or $1$ as predicting that $x$ is generated from class $\mu_0$ or $\mu_1$. We view $\mu_0$ and $\mu_1$ as representing a *common* and *rare* class of (future) data.

**Definition 2.1.** *We are interested in the so-called* Type-I *and* Type-II *errors defined as follows:*
$$R_{\mu_0}(h) = \mu_0(h(x) = 1), \quad R_{\mu_1}(h) = \mu_1(h(x) = 0).$$

*Neyman-Pearson* classification then refers to the problem of minimizing Type-II error subject to low Type-I error:
$$\underset{h \in \mathcal{H}}{\text{Minimize}}\ R_{\mu_1}(h)$$
$$\text{s.t. } R_{\mu_0}(h) \leq \alpha \tag{2.1}$$

Under mild conditions, the *universally* optimal classifier, i.e., taking $\mathcal{H}$ as the set of all measurable 0-1 functions, is fully characterized by the classical Neyman-Pearson Lemma (see Appendix A) in terms of *density ratios*. Namely, let $p_0$ and $p_1$ denote densities of $\mu_0$ and $\mu_1$ w.r.t. some dominating measure $\nu$, then the minimizer of (2.1) has the form $h_\alpha^*(x) = \mathbb{1}_{\left\{\frac{p_1(x)}{p_0(x)} \geq \lambda\right\}}$ whenever there exists $\lambda$ such that $R_{\mu_0}(h_\alpha^*)$ is exactly $\alpha$[1].

---

[1] If we further allow for randomized classifiers, then Neyman Pearson Lemma fully characterizes universally optimal solutions of (2.1) and establishes uniqueness almost-surely under mild restrictions.

In the Section 3 we ask when such universal minimizer transfers across source and target rare-class distributions.

## 2.2 TRANSFER LEARNING SETUP

**Population Setup.** We consider the following two source and target Neyman-Pearson problems, defined for a fixed common class distribution $\mu_0$, and source and target rare-class distributions $\mu_{1,S}$ and $\mu_{1,T}$:

$$\underset{h \in \mathcal{H}}{\text{Minimize }} R_{\mu_{1,S}}(h) \qquad\qquad\qquad \underset{h \in \mathcal{H}}{\text{Minimize }} R_{\mu_{1,T}}(h)$$
$$\text{s.t. } R_{\mu_0}(h) \leq \alpha \qquad (2.2) \qquad\qquad \text{s.t. } R_{\mu_0}(h) \leq \alpha \qquad (2.3)$$

We let $(\mu_0, \mu_{1,S}, \alpha)$ and $(\mu_0, \mu_{1,T}, \alpha)$ denote these source and target problems. We will see later that the limits of outlier-transfer, especially in finite-sample regimes, are well captured by discrepancies between these two problems. In particular, we will be interested in discrepancies between optimal solutions and the measure of the corresponding decision boundaries under involved distributions. We henceforth let $h^*_{S,\alpha}$ and $h^*_{T,\alpha}$ denote (not necessarily unique) **solutions** of (2.2) and (2.3) (which we assume exist).

**Finite-Sample Setup.** We assume access to $n_0, n_S, n_T$ i.i.d. data points respectively from $\mu_0, \mu_{1,S}, \mu_{1,T}$, where we allow $n_T = 0$. The transfer-learning procedure is then allowed to return $\hat{h} \in \mathcal{H}$ satisfying

$$R_{\mu_0}(\hat{h}) \leq \alpha + \epsilon_0,$$

for some slack $\epsilon_0 = \epsilon_0(n_0)$, usually of order $n_0^{-1/2}$. The goal of the learner is to minimize the **target-excess error**

$$\mathcal{E}_{1,T}(\hat{h}) \doteq \max\left\{0, R_{\mu_{1,T}}(\hat{h}) - R_{\mu_{1,T}}(h^*_{T,\alpha})\right\}.$$

A main aim of this work is to understand which rates of $\mathcal{E}_{1,T}(\hat{h})$ are achievable in terms of sample sizes $n_S$ and $n_T$, and which notions of discrepancy from source to target helps capture these limits.

## 3 EQUIVALENCE OF POPULATION PROBLEMS

As discussed in the introduction, we may have seemingly very different source and target distributions $\mu_{1,S}$ and $\mu_{1,T}$ which however yield the same (universally) optimal classifiers. We investigate this phenomena in this section, the main aim being to illustrate how fundamentally different outlier transfer is from traditional classification. To this end we consider the set $\mathcal{U}$ of all possible measurable 0-1 functions on $\mathcal{X}$, and let $\mathcal{H} = \mathcal{U}$ in the dicussion below.

We will henceforth say that the source problem $(\mu_0, \mu_{1,S}, \alpha)$ is **equivalent** to the target problem $(\mu_0, \mu_{1,T}, \alpha)$ (at the population level), if all solutions to (2.2) are also solutions to (2.3). Clearly, when this is the case, source samples alone can drive down the target risk at least asymptotically.

We first notice that Neyman-Pearson Lemma offers some immediate answers under mild conditions. To see this, let $p_0, p_{1,S}, p_{1,T}$ denote densities w.r.t. a common dominating measure $\nu$. In what follows we will simply let the dominating measure $\nu \doteq \mu_0 + \mu_{1,S} + \mu_{1,T}$. As previously discussed, Neyman-Pearson Lemma characterizes optimal classifiers in terms of level sets of density ratios.

**Definition 3.1** (Level sets). $\mathcal{L}^S_\lambda \coloneqq \{x : \frac{p_{1,S}(x)}{p_0(x)} \geq \lambda\}$ and $\mathcal{L}^T_\lambda \coloneqq \{x : \frac{p_{1,T}(x)}{p_0(x)} \geq \lambda\}$. *Here, when the denominator of a fraction is zero we view it as infinity no matter if the nominator is zero or nonzero.*

The following then establishes *equivalence* between source and target under mild conditions as a direct consequence of Neyman-Pearson Lemma.

**Proposition 3.2.** *Suppose $\mu_0, \mu_{1,S}, \mu_{1,T}$ are mutually dominating. Assume further that $\mu_0(\mathcal{L}^S_\lambda)$ is continuous and strictly monotonic in $(0,1)$ as a function of $\lambda$. Then, if $\{\mathcal{L}^S_\lambda\}_{\lambda \geq 0} \subset \{\mathcal{L}^T_\lambda\}_{\lambda \geq 0}$, we have for all $0 < \alpha < 1$, that any $h^*_{S,\alpha}$ is a solution of the target problem (2.3).*

The above statement was already evident in the work of Scott (2019) where it is assumed that source density ratios are given as a monotonic function of the target density ratios; this immediately implies $\{\mathcal{L}^S_\lambda\} \subset \{\mathcal{L}^T_\lambda\}$.

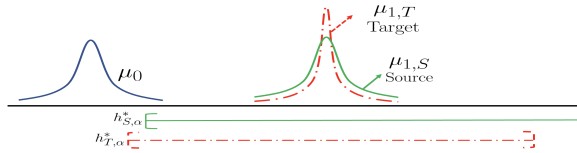

Figure 2: $\mu_0, \mu_{1,S}, \mu_{1,T} = \mathcal{N}(a_0, \sigma^2), \mathcal{N}(a_{1,S}, \sigma^2), \mathcal{N}(a_{1,T}, \sigma'^2)$ where $a_{1,S} = a_{1,T}$ and $\sigma' < \sigma$. Optimal decision rules differ.

The statement is illustrated in the examples of Fig 1 with Gaussian distributions where the source may appear significantly different from the target (and in fact would yield different Bayes classifiers in traditional classification). To contrast, consider the example Fig 2 where the source problem yields different level sets than for the target problem (simply by changing the Gaussian variance), and we hence do not have equivalence.

**Remark 3.1** (Issue with Disjoint Supports). *We have assumed in the Proposition 3.2 that all 3 distributions are mutually dominating, while in practice this might rarely be the case. However, without this assumption (essentially of shared support), we may not easily have equivalence between source and target.*

*For intuition, consider a region $A$ of space where $\mu_{1,S}(A) = 0$ while $\mu_{1,T}(A) > 0$. Then let $h_{S,\alpha}^{*,0} = 0$ on $A$ and $h_{S,\alpha}^{*,1} = 1$ both optimal under the source problem; clearly we may have $R_{1,T}(h_{S,\alpha}^{*,0}) > R_{1,T}(h_{S,\alpha}^{*,1})$ since $\mu_{1,T}(A) > 0$.*

The rest of this section is devoted to handling this issue by restricting attention to more *reasonable* classifiers that essentially classify any point outside the support of $\mu_0$ as 1. The right notion of support is critical in establishing our main relaxation of the above proposition, namely Proposition 3.6 below.

We require the following definitions and assumptions.

**Definition 3.3** (Restricted Universal Hyposthesis Class). *We let $\mathcal{U}^*$ consist of all 0-1 measurable functions on the domain $\mathcal{X}$ such that for every $h \in \mathcal{U}^*$, $h \equiv 1$ on $\{x : p_0(x) = 0\}$ a.s. $\nu$.*

**Definition 3.4.** *We say that $\alpha$ is achievable if there exist thresholds $\lambda$ and $\lambda'$ such that $\mu_0(\mathcal{L}_\lambda^S) = \alpha$ and $\mu_0(\mathcal{L}_{\lambda'}^T) = \alpha$.*

**Definition 3.5** ($\alpha$-level set). *Whenever $\alpha$ is achievable, we define $\mathcal{L}^S(\alpha)$ as the level set in the source whose measure under $\mu_0$ is $\alpha$. The definition of $\mathcal{L}^T(\alpha)$ which corresponds to the target is the same.*

**Remark 3.2.** *Definition 3.4 ensures that $\mathcal{L}^S(\alpha)$ exists, but it may not be unique. However, we will show in Appendix A by Proposition A.2, it is unique a.s. $\nu$.*

The following proposition relaxes the conditions of Proposition 3.2 above by restricting attention to universal classifiers in $\mathcal{U}^*$. Its proof is technical to various corner cases described in the Appendix.

**Proposition 3.6.** *Let $0 \le \alpha < 1$ and suppose that $\alpha$ is achievable. Then $(\mu_0, \mu_{1,S}, \alpha)$ is equivalent to $(\mu_0, \mu_{1,T}, \alpha)$ under $\mathcal{U}^*$ iff $\mathcal{L}^S(\alpha) \in \{\mathcal{L}_\lambda^T\}_{\lambda \ge 0}$ a.s. $\nu$. In particular, if $\alpha$ is achievable for all $0 \le \alpha < 1$ and $\mathcal{L}^S(\alpha) \in \{\mathcal{L}_\lambda^T\}_{\lambda \ge 0}$ a.s. $\nu$ for all $0 \le \alpha < 1$, then $(\mu_0, \mu_{1,S}, \alpha)$ is equivalent to $(\mu_0, \mu_{1,T}, \alpha)$ for all $0 \le \alpha < 1$.*

**Remark 3.3.** *Notice that the statements of Proposition 3.6 trivially also hold over any hypothesis class $\mathcal{H}$ (rather than just $\mathcal{U}^*$) containing the level sets $\mathbb{1}_{\mathcal{L}^S(\alpha)}, \mathbb{1}_{\mathcal{L}^T(\alpha)} \in \mathcal{H}$ and where, for every $h \in \mathcal{H}$, $h \equiv 1$ on $\{x : p_0(x) = 0\}$ a.s. $\nu$.*

We illustrate this final Proposition in Figure 3: the simple restriction to $\mathcal{U}^*$ reveals more scenarios where source is equivalent to target (at the population level) even when the distributions appear significantly different.

## 4 FINITE SAMPLE RESULTS

Neyman-Pearson Lemma offers the solution(s) of the optimization problem (2.3) when we have the knowledge of the underlying distributions. However, in practical scenarios, we typically lack infor-

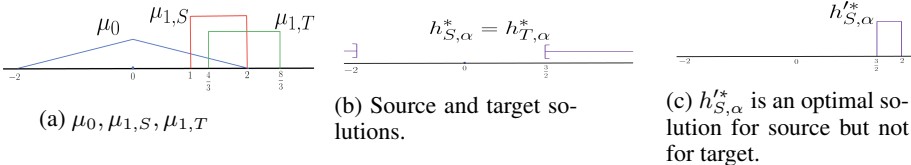

(a) $\mu_0, \mu_{1,S}, \mu_{1,T}$

(b) Source and target solutions.

(c) $h'^*_{S,\alpha}$ is an optimal solution for source but not for target.

Figure 3: Illustration of Example A.4 (see Appendix A.3), where $h^*_{S,\alpha} = h^*_{T,\alpha}$ in $\mathcal{U}^*$, while $h'^*_{S,\alpha} \in \mathcal{U}$ is also optimal for source but not for target. In other words, the source problem remains equivalent to the target over the more *reasonable* decision rules of $\mathcal{U}^*$.

mation about these distributions and only possess some samples drawn from them. In addressing this challenge, Cannon et al. (2002) embarked on an empirical study of Neyman-Pearson classification and introduced a relaxed version of Neyman-Pearson classification problem. Let $n_0$ and $n_T$ be the number of i.i.d. samples generated by $\mu_0$ and $\mu_{1,T}$, respectively, and $\epsilon_0 > 0$. Cannon et al. (2002) proposed the following optimization problem:

$$\hat{h} = \arg\min_{h \in \mathcal{H}} \hat{R}_{\mu_{1,T}}(h)$$

$$\text{s.t. } \hat{R}_{\mu_0}(h) \leq \alpha + \frac{\epsilon_0}{2} \tag{4.1}$$

where $\hat{R}_{\mu_0}(h) = \frac{1}{n_0} \sum_{X_i \sim \mu_0} \mathbb{1}_{\{h(X_i) \neq 0\}}$ and $\hat{R}_{\mu_{1,T}}(h) = \frac{1}{n_T} \sum_{X_i \sim \mu_{1,T}} \mathbb{1}_{\{h(X_i) \neq 1\}}$, and then derived the convergence rate of excess error in terms of the number of samples and VC dimension of the hypothesis class.

Neyman-Pearson classification in the setting of transfer finite-sample scenarios remains unexplored. In this section, Our objective is to understand the fundamental limits of transfer outlier detection in the finite-sample regime, where there are $n_0, n_S, n_T$ i.i.d. samples from $\mu_0, \mu_{1,S}, \mu_{1,T}$, under a measure of discrepancy between source and target. We first define a discrepancy measure in transfer outlier detection and then characterize the fundamental limits of the problem by deriving a minimax lower bound in terms of the number of samples as well as the notion of discrepancy. Finally, we show that this lower bound is achievable through an adaptive procedure which does not require the prior knowledge of the discrepancy between source and target.

## 4.1 OUTLIER TRANSFER EXPONENT

In this section, we define an appropriate notion of outlier transfer distance between source and target under a hypothesis class. Here we adapt the transfer exponent notion defined in Hanneke & Kpotufe (2019) to a notion of discrepancy between source and target in Neyman-Pearson classification with shared common distribution $\mu_0$.

**Definition 4.1** (Outlier transfer exponent). *Let $S^*_\alpha \subset \mathcal{H}$ denote the set of solutions of source (2.2). Furthermore, let $(\mu_0, \mu_{1,S}, \alpha)$ and $(\mu_0, \mu_{1,T}, \alpha)$ denote the source and target problems, respectively. We call $\rho(r) > 0$ outlier transfer exponent from $(\mu_0, \mu_{1,S}, \alpha)$ to $(\mu_0, \mu_{1,T}, \alpha)$ under $\mathcal{H}$, if there exist $r, C_{\rho(r)} > 0$ such that*

$$C_{\rho(r)} \cdot \max\left\{0, R_{\mu_{1,S}}(h) - R_{\mu_{1,S}}(h^*_{S,\alpha})\right\} \geq \max\left\{0, R_{\mu_{1,T}}(h) - R_{\mu_{1,T}}(h^*_{S,\alpha})\right\}^{\rho(r)} \tag{4.2}$$

*for all $h \in \mathcal{H}$ with $R_{\mu_0}(h) \leq \alpha + r$, where $h^*_{S,\alpha} = \arg\max_{h \in S^*_\alpha} R_{\mu_{1,T}}(h)$.*

The following example shows that for a fixed target and for any arbitrary $\rho \geq 1$, there exists a source such that it shares the same optimal decision rule as the target and attains the outlier transfer exponent $\rho$ with coefficient $C_\rho = 1$.

**Example 4.2.** *Let $\mu_0 \sim \mathcal{N}(-1, 1)$, $\mu_{1,T} \sim Unif[0, 1]$, $p_{1,S} = \rho x^{\rho-1} \mathbb{1}_{\{x \in [0,1]\}}$ for $\rho \geq 1$ where $p_{1,S}$ is the density of $\mu_{1,S}$ w.r.t. Lebesgue measure. Furthermore, let $\alpha = \mu_0([0, 1])$, $\mathcal{H} = \{\mathbb{1}_{\{t \leq x \leq 1\}}(x) : t \in$*

$[-1, 1]\}$, *and* $r$ *be small enough. Then, we have* $h^*_{T,\alpha} = h^*_{S,\alpha} = \mathbb{1}_{\{0 \le x \le 1\}}$. *Moreover, for* $h = \mathbb{1}_{\{t \le x \le 1\}}$
*for* $t \ge 0$ *we obtain*

$$R_{\mu_{1,T}}(h) - R_{\mu_{1,T}}(h^*_{S,\alpha}) = t \quad and \quad R_{\mu_{1,S}}(h) - R_{\mu_{1,S}}(h^*_{S,\alpha}) = t^\rho.$$

*Hence, the outlier transfer exponent is* $\rho$ *and the coefficient* $C_\rho$ *is* 1.

Following proposition shows the effect of $r$ on the outlier transfer exponent $\rho(r)$. In some cases, for a small enough value of $r$, $\rho(r)$ could be small, whereas for a large value of $r$, $\rho(r)$ is infinite.

**Proposition 4.2.** *There exist* $\mu_0, \mu_{1,S}, \mu_{1,T}, \mathcal{H}, \alpha, r$ *such that for any* $h \in \mathcal{H}$ *with* $R_{\mu_0}(h) \le \alpha + r$, *(4.2) holds for* $\rho(r) = 1$ *and there exists an* $h \in \mathcal{H}$ *with* $R_{\mu_0}(h) > \alpha + r$ *for which (4.2) does not hold for any* $\rho < \infty$.

### 4.3 MINIMAX LOWER BOUND FOR OUTLIER TRANSFER LEARNING

Equipped with the notion of outlier transfer exponent, we characterize the fundamental limits of transfer outlier detection by establishing a minimax lower bound. To achieve this objective, first we need to specify the class of distributions for which the minimax lower bound is derived.

**Definition 4.3** (Class of distributions). *Fix a hypothesis class* $\mathcal{H}$ *with finite VC dimension* $d_\mathcal{H}$. *Let* $\mathcal{F}_\mathcal{H}(\rho, \alpha, C, \Delta)$ *denote the class of triplets of distributions* $(\mu_0, \mu_{1,S}, \mu_{1,T})$ *for which* $\alpha$ *is achievable according to Definition 3.4,* $\mathcal{E}_{1,T}(h^*_{S,\alpha}) \le \Delta$, *and there exist* $\rho(r) \le \rho, C_{\rho(r)} \le C$ *for any* $0 < r < \frac{2\alpha}{d_\mathcal{H}}$ *according to Definition 4.1.*

**Remark 4.4.** *Deriving a minimax lower bound for the class of distributions satisfying* $\alpha$ *achievability is a stronger result than for the class without necessarily satisfying that, as the former is contained in the latter.*

We also need to formalize the class of learners.

**Definition 4.4.** *Let* $0 \le \epsilon_0 < 1$, $0 < \delta_0 < 1$. *We call* $\hat{h}$ *an* $(\epsilon_0, \delta_0)$**-approximate** $\alpha$**-learner** *if it maps any three independent i.i.d. samples* $S_{\mu_0} \sim \mu_0^{n_0}$, $S_{\mu_{1,S}} \sim \mu_{1,S}^{n_S}$, $S_{\mu_{1,T}} \sim \mu_{1,T}^{n_T}$, *to a function in*

$$\mathcal{H}_{\alpha+\epsilon_0}(\mu_0) = \{h \in \mathcal{H} : R_{\mu_0}(h) \le \alpha + \epsilon_0\}$$

*with probability at least* $1 - \delta_0$ *w.r.t.* $S_{\mu_0}, S_{\mu_{1,S}}, S_{\mu_{1,T}}$.

**Theorem 4.5** (Minimax lower bound). *Fix a hypothesis class* $\mathcal{H}$ *with finite VC dimension* $d_\mathcal{H} \ge 3$. *Moreover, let* $\alpha < \frac{1}{2}, \rho \ge 1, \Delta \le 1$, *and* $\delta_0 > 0$. *Furthermore, suppose that there are* $n_0, n_S, n_T$ *i.i.d. samples from* $\mu_0, \mu_{1,S}, \mu_{1,T}$, *denoted by* $S_{\mu_0}, S_{\mu_{1,S}}, S_{\mu_{1,T}}$. *Assume that there are sufficiently many samples such that* $\min\{\Delta + (\frac{d_\mathcal{H}}{n_S})^{\frac{1}{2\rho}}, (\frac{d_\mathcal{H}}{n_T})^{\frac{1}{2}}\} \le 2$. *Then, for any* $(\frac{2\alpha}{d_\mathcal{H}}, \delta_0)$-*approximate* $\alpha$-*learner* $\hat{h}$, *there exist* $(\mu_0, \mu_{1,S}, \mu_{1,T}) \in \mathcal{F}_\mathcal{H}(\rho, \alpha, 1, \Delta)$ *and universal constants* $c, c'$ *such that*

$$\mathbb{P}_{S_{\mu_0}, S_{\mu_{1,S}}, S_{\mu_{1,T}}} \left( \mathcal{E}_{1,T}(\hat{h}) > c \cdot \min \left\{ \Delta + (\frac{d_\mathcal{H}}{n_S})^{\frac{1}{2\rho}}, (\frac{d_\mathcal{H}}{n_T})^{\frac{1}{2}} \right\} \right) > c'. \tag{4.3}$$

**Remark 4.5.** *In Theorem 4.5, if* $(\mu_0, \mu_{1,S}, \alpha)$ *is equivalent to* $(\mu_0, \mu_{1,T}, \alpha)$ *under* $\mathcal{H}$ *(see Remark 3.3), then* $\mathcal{E}_{1,T}(h^*_{S,\alpha}) = 0$ *and the minimax lower bound reduces to* $c \cdot \min\{(\frac{d_\mathcal{H}}{n_S})^{\frac{1}{2\rho}}, (\frac{d_\mathcal{H}}{n_T})^{\frac{1}{2}}\}$. *In this case, by only having access to unlimited source samples, achieving an arbitrary small target-excess error is possible.*

**Remark 4.6.** *The outlier transfer exponent in the term* $(\frac{d_\mathcal{H}}{n_S})^{\frac{1}{2\rho}}$ *captures the relative effectiveness of source samples in the target domain. If source and target share the same optimal decision rules, and* $\rho = 1$, *source samples would be equally effective as target samples. However, even if the source and target share the same optimal decision rules, source samples may result in poor transfer performance when* $\rho$ *is large.*

**Remark 4.7.** *In Theorem 4.5, the learner is allowed to output a classifier* $\hat{h}$ *with a Type-I error that slightly exceeds the pre-defined threshold* $\alpha$. *However, in certain applications, it is imperative to uphold the threshold without any exceeding. The minimax lower bound in Theorem 4.5, implies that (4.3) holds even if the learner is only allowed to output a classifier* $\hat{h}$ *from* $\{h \in \mathcal{H} : R_{\mu_0}(h) \le \alpha\}$ *with probability* $1 - \delta_0$, *i.e.,* $(0, \delta_0)$-*approximate* $\alpha$-*learner.*

### 4.8 ADAPTIVE RATE (UPPER BOUND)

In Section 4.3, we establish a minimax lower bound that can be attained by an oracle that effectively disregards the least informative dataset from either the source or target. Let $\hat{\mathcal{H}}_{\alpha+\epsilon_0}(\mu_0) = \{h \in \mathcal{H} : \hat{R}_{\mu_0}(h) \le \alpha + \epsilon_0\}$, $\hat{h}_T = \underset{h \in \hat{\mathcal{H}}_{\alpha+\epsilon_0/2}(\mu_0)}{\arg\min} \hat{R}_{\mu_{1,T}}(h)$, and $\hat{h}_S = \underset{h \in \hat{\mathcal{H}}_{\alpha+\epsilon_0/2}(\mu_0)}{\arg\min} \hat{R}_{\mu_{1,S}}(h)$. Then we get $\hat{h}_S, \hat{h}_T \in \mathcal{H}_{\alpha+\epsilon_0}(\mu_0)$, and

$$\mathcal{E}_{1,T}(\hat{h}_S) \le \mathcal{E}_{1,T}(h_{S,\alpha}^*) + \left(\frac{d_{\mathcal{H}}}{n_S}\right)^{\frac{1}{2\rho}} \text{ and } \mathcal{E}_{1,T}(\hat{h}_T) \le \left(\frac{d_{\mathcal{H}}}{n_T}\right)^{\frac{1}{2}}$$

with high probability. However, Deciding whether $\hat{h}_S$ or $\hat{h}_T$ achieves a smaller error requires the knowledge of outlier transfer exponent and the target-excess error of the optimal source decision rule, which are not available in practical scenarios.

In this section, we show that by using an adaptive procedure that takes source and target samples as input and produces a hypothesis $\hat{h} \in \mathcal{H}$ without using any additional information such as prior knowledge of the outlier transfer exponent, the minimax lower bound 4.3 is achievable. To accomplish this, we adapt the procedure introduced in Hanneke & Kpotufe (2019). Let $\delta > 0$ and define

$$A_n = \sqrt{128\frac{d_{\mathcal{H}}\log n + \log(8/\delta)}{n}}.$$

Consider the following procedure:

Define $\hat{h} = \hat{h}_S$ if $\hat{R}_{\mu_{1,T}}(\hat{h}_S) - \hat{R}_{\mu_{1,T}}(\hat{h}_T) \le A_{n_T}$,

otherwise, define $\hat{h} = \hat{h}_T$                            (4.4)

**Theorem 4.6.** *Let $\mathcal{H}$ be a hypothesis class with finite VC dimension $d_{\mathcal{H}} \ge 3$. Furthermore, let $(\mu_0, \mu_{1,S}, \alpha)$ and $(\mu_0, \mu_{1,T}, \alpha)$ denote a source and a target problem. Suppose that there are $n_0, n_S, n_T$ i.i.d. samples from $\mu_0, \mu_{1,S}, \mu_{1,T}$, respectively. Let $\delta_0, \delta > 0$, $\epsilon_0 = \sqrt{128\frac{d_{\mathcal{H}}\log n_0 + \log(8/\delta_0)}{n_0}}$. Moreover, suppose that there exist $r \ge \epsilon_0$, $C_{\rho(r)}$, and $\rho(r)$ according to Definition 4.1. Let $\hat{h}$ be the hypothesis returned by Procedure (4.4). Then, with probability at least $1 - \delta_0$, $\hat{h} \in \mathcal{H}_{\alpha+\epsilon_0}(\mu_0)$. Furthermore, with probability at least $1 - \delta_0 - 2\delta$ we have*

$$\mathcal{E}_{1,T}(\hat{h}) \le \min\left\{\mathcal{E}_{1,T}(h_{S,\alpha}^*) + C \cdot A_{n_S}^{\frac{1}{\rho(r)}}, C \cdot A_{n_T}\right\} \tag{4.5}$$

*where $C \in (0, \infty)$ is a constant depending on $r, C_{\rho(r)}, \rho(r)$.*

*Proof.* Consider the intersection of the following events which happens with probability at least $1 - \delta_0 - 2\delta$ (Devroye et al., 2013): 1) $\{\sup_{h \in \mathcal{H}}|R_{\mu_0}(h) - \hat{R}_{\mu_0}(h)| \le \frac{\epsilon_0}{2}\}$, 2) $\{\sup_{h \in \mathcal{H}}|R_{\mu_{1,S}}(h) - \hat{R}_{\mu_{1,S}}(h)| \le \frac{A_{n_S}}{2}\}$, and 3) $\{\sup_{h \in \mathcal{H}}|R_{\mu_{1,T}}(h) - \hat{R}_{\mu_{1,T}}(h)| \le \frac{A_{n_T}}{2}\}$. Since $h_{S,\alpha}^* \in \hat{\mathcal{H}}_{\alpha+\epsilon_0/2}(\mu_0)$, we get $\hat{R}_{\mu_{1,S}}(\hat{h}_S) \le \hat{R}_{\mu_{1,S}}(h_{S,\alpha}^*)$. Therefore, we obtain

$$R_{\mu_{1,S}}(\hat{h}_S) - R_{\mu_{1,S}}(h_{S,\alpha}^*) \le \left[R_{\mu_{1,S}}(\hat{h}_S) - \hat{R}_{\mu_{1,S}}(\hat{h}_S)\right] + \left[\hat{R}_{\mu_{1,S}}(h_{S,\alpha}^*) - R_{\mu_{1,S}}(h_{S,\alpha}^*)\right] \le A_{n_S}.$$

Furthermore, we have $R_{\mu_0}(\hat{h}_S) \le \alpha + r$, which implies that

$$R_{\mu_{1,T}}(\hat{h}_S) - R_{\mu_{1,T}}(h_{S,\alpha}^*) \le c \cdot A_{n_S}^{\frac{1}{\rho(r)}}.$$

Hence,

$$R_{\mu_{1,T}}(\hat{h}_S) - R_{\mu_{1,T}}(h_{T,\alpha}^*) = R_{\mu_{1_T}}(\hat{h}_S) - R_{\mu_{1,T}}(h_{S,\alpha}^*) + R_{\mu_{1,T}}(h_{S,\alpha}^*) - R_{\mu_{1,T}}(h_{T,\alpha}^*)$$

$$\le \mathcal{E}_{1,T}(h_{S,\alpha}^*) + c \cdot A_{n_S}^{\frac{1}{\rho(r)}}. \tag{4.6}$$

Now if $R_{\mu_{1,T}}(\hat{h}_S) \le R_{\mu_{1,T}}(\hat{h}_T)$, we get

$$\hat{R}_{\mu_{1,T}}(\hat{h}_S) - \hat{R}_{\mu_{1,T}}(\hat{h}_T) \le \hat{R}_{\mu_{1,T}}(\hat{h}_S) - R_{\mu_{1,T}}(\hat{h}_S) + R_{\mu_{1,T}}(\hat{h}_T) - \hat{R}_{\mu_{1,T}}(\hat{h}_T) \le A_{n_T}$$

which shows that the constraint in Procedure (4.4) holds, implying that $\hat{h} = \hat{h}_S$ and the upper bound (4.6) holds for $R_{\mu_{1,T}}(\hat{h}) - R_{\mu_{1,T}}(h^*_{T,\alpha})$. On the other hand, if $R_{\mu_{1,T}}(\hat{h}_S) > R_{\mu_{1,T}}(\hat{h}_T)$, then

$$R_{\mu_{1,T}}(\hat{h}_T) - R_{\mu_{1,T}}(h^*_{T,\alpha}) < R_{\mu_{1,T}}(\hat{h}_S) - R_{\mu_{1,T}}(h^*_{T,\alpha}).$$

So, regardless of whether $\hat{h} = \hat{h}_S$ or $\hat{h} = \hat{h}_T$, the upper bound (4.6) holds for $R_{\mu_{1,T}}(\hat{h}) - R_{\mu_{1,T}}(h^*_{T,\alpha})$. Moreover, Since $\hat{h}$ satisfies the constraint in Procedure (4.4), we get

$$R_{\mu_{1,T}}(\hat{h}) - R_{\mu_{1,T}}(h^*_{T,\alpha}) = R_{\mu_{1,T}}(\hat{h}) - R_{\mu_{1,T}}(\hat{h}_T) + R_{\mu_{1,T}}(\hat{h}_T) - R_{\mu_{1,T}}(h^*_{T,\alpha}) \le 3\epsilon_T$$

which completes the proof. $\qquad\square$

## 5   Overview of Proof of Theorem 4.5 (Minimax Lower Bound)

We follow Tysbakov's method (Tsybakov, 2009) for derivng the minimax lower bound.

**Theorem 5.1.** *Tsybakov (2009) Assume that $M \ge 2$ and the function $dist(\cdot, \cdot)$ is a semi-metric. Furthermore, suppose that $\{\Pi_{\theta_j}\}_{\theta_j \in \Theta}$ is a family of distributions indexed over a parameter space, $\Theta$, and $\Theta$ contains elements $\theta_0, \theta_1, ..., \theta_M$ such that:*

    *(i) $dist(\theta_i, \theta_j) \ge 2s > 0, \quad \forall\ 0 \le i < j \le M$*

    *(ii) $\Pi_j \ll \Pi_0, \quad \forall\ j = 1, ..., M$, and $\frac{1}{M}\sum_{j=1}^M \mathcal{D}_{kl}(\Pi_j | \Pi_0) \le \gamma \log M$ with $0 < \gamma < 1/8$ and $\Pi_j = \Pi_{\theta_j}, j = 0, 1, ..., M$ and $\mathcal{D}_{kl}$ denotes the KL-divergence.*

*Then, we have $\inf_{\hat{\theta}} \sup_{\theta \in \Theta} \Pi_\theta(dist(\hat{\theta}, \theta) \ge s) \ge \frac{\sqrt{M}}{1+\sqrt{M}}\left(1 - 2\gamma - \sqrt{\frac{2\gamma}{\log M}}\right).$*

We also utilize the following proposition for constructing a packing of the parameter space.

**Proposition 5.2.** *(Gilbert-Varshamov bound) Let $d \ge 8$. Then there exists a subset $\{\sigma_0, ..., \sigma_M\}$ of $\{-1, +1\}^d$ such that $\sigma_0 = (1, 1, ..., 1)$,*

$$dist(\sigma_j, \sigma_k) \ge \frac{d}{8}, \quad \forall\ 0 \le j < k \le M \text{ and } M \ge 2^{d/8},$$

*where $dist(\sigma, \sigma') = card(i \in [m] : \sigma(i) \ne \sigma'(i))$ is the Hamming distance.*

First note that

$$\min\{\Delta + (\frac{d_{\mathcal{H}}}{n_S})^{\frac{1}{2\rho}}, (\frac{d_{\mathcal{H}}}{n_T})^{\frac{1}{2}}\} \le 2 \cdot \min\{\max\{\Delta, (\frac{d_{\mathcal{H}}}{n_S})^{\frac{1}{2\rho}}\}, (\frac{d_{\mathcal{H}}}{n_T})^{\frac{1}{2}}\}$$

$$= 2 \cdot \max\{\min\{(\frac{d_{\mathcal{H}}}{n_S})^{\frac{1}{2\rho}}, (\frac{d_{\mathcal{H}}}{n_T})^{\frac{1}{2}}\}, \min\{\Delta, (\frac{d_{\mathcal{H}}}{n_T})^{\frac{1}{2}}\}\}$$

So it suffices to show that the minimax lower bound is larger than both $\min\{(\frac{d_{\mathcal{H}}}{n_S})^{\frac{1}{2\rho}}, (\frac{d_{\mathcal{H}}}{n_T})^{\frac{1}{2}}\}$ and $\min\{\Delta, (\frac{d_{\mathcal{H}}}{n_T})^{\frac{1}{2}}\}$. We divide the proof into three parts:

- Minimax lower bound is larger than $\min\{(\frac{d_{\mathcal{H}}}{n_S})^{\frac{1}{2\rho}}, (\frac{d_{\mathcal{H}}}{n_T})^{\frac{1}{2}}\}$ for $d_{\mathcal{H}} \ge 17$ (see Section C.1).

- Minimax lower bound is larger than $\min\{(\frac{d_{\mathcal{H}}}{n_S})^{\frac{1}{2\rho}}, (\frac{d_{\mathcal{H}}}{n_T})^{\frac{1}{2}}\}$ for $16 \ge d_{\mathcal{H}} \ge 3$ (see Section C.2).

- Minimax lower bound is larger than $\min\{\Delta, (\frac{d_{\mathcal{H}}}{n_T})^{\frac{1}{2}}\}$ (see Section C.3).

In each part, Following Theorem 5.1, we construct a family of pairs of source and target distributions that belong to the class $\mathcal{F}_{\mathcal{H}}$. To accomplish this, we pick some points from the domain $\mathcal{X}$ shattered by the hypothesis class $\mathcal{H}$ and then define appropriate distributions on these points. Additionally, this family of distributions is indexed by $\{-1, +1\}^{d_{\mathcal{H}}}$, which can be treated as a metric space using the Hamming distance. To meet the requirement of condition (i) in Theorem 5.1, it is necessary for these indices to be well-separated, a condition that can be satisfied through utilizing Proposition 5.2.

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

## A   APPENDIX A (EQUIVALENCE OF POPULATION PROBLEMS)

We begin by stating Neyman-Pearson lemma (Lehmann & Lehmann, 1986) for deterministic tests. In the following, a classifier $h : \mathcal{X} \to \{0, 1\}$ aims at classifying $H_0 : \mu_0$ against the alternative $H_1 : \mu_1$. In the context of hypothesis testing, $h$ is called a deterministic test. Moreover, $R_{\mu_0}(h)$ and $1 - R_{\mu_1}(h)$ are called *size* and *power*, respectively.

**Theorem A.1** (Neyman-Pearson Lemma (Lehmann & Lehmann, 1986)). *Let $\mu_0$ and $\mu_1$ be probability distributions possessing densities $p_0$ and $p_1$ respectively with respect to a dominating measure $\nu$.*

*(i) Sufficient condition for an optimal solution of the optimization problem (2.1). Let $h$ be a classifier for $H_0 : \mu_0$ against the alternative $H_1 : \mu_1$ such that for a constant $\lambda$ the followings hold*

$$R_{\mu_0}(h) = \alpha \tag{A.1}$$

*and*

$$h(x) = \begin{cases} 1 & when \quad p_1(x) \geq \lambda p_0(x) \\ 0 & when \quad p_1(x) < \lambda p_0(x) \end{cases} \tag{A.2}$$

*Then $h$ is an optimal solution of (2.1).*

*(ii) Necessary condition for an optimal solution of the optimization problem (2.1). Suppose that there exist a classifier $h$ and a constant $\lambda$ such that (A.1) and (A.2) hold. Then, any solution of (2.1), denoted by $h^*$, satisfies the following a.s. $\nu$*

$$h^*(x) = \begin{cases} 1 & when \quad p_1(x) > \lambda p_0(x) \\ 0 & when \quad p_1(x) < \lambda p_0(x) \end{cases} \tag{A.3}$$

$h^*$ *also satisfies* $R_{\mu_0}(h^*) = \alpha$ *unless there exists a classifier $h'$ with $R_{\mu_0}(h') < \alpha$ and $R_{\mu_1}(h') = 0$.*

**Remark A.1.** *To obtain a solution of the optimization problem (2.2) in the source (or (2.3) in the target) it suffices to take the level set $\mathcal{L}_\lambda^S$ ($\mathcal{L}_{\lambda'}^T$ in the target) whose measure under $\mu_0$ is $\alpha$, and then define a classifier $h$ as $h = \mathbb{1}_{\mathcal{L}_\lambda^S}$ in the source ($h = \mathbb{1}_{\mathcal{L}_{\lambda'}^T}$ in the target). Obviously, the classifier $h$ satisfies (A.1) and (A.2), and therefore it is an optimal solution.*

**Proposition A.2.** *Suppose that there exist a classifier $h$ with $h \equiv 1$ on $\{x : p_0(x) = 0\}$ a.s. $\nu$ and a constant $\lambda$ such that (A.1) and (A.2) hold for $h$. Then any solution of (2.1), denoted by $h^*$, such that $h^* \equiv 1$ on $\{x : p_0(x) = 0\}$ a.s. $\nu$ and $R_{\mu_0}(h^*) = \alpha$ satisfies (A.2) a.s. $\nu$.*

*Proof.* Let $S_1 = \{x : h^*(x) = h(x)\}$, $S_2 = \{x : h^*(x) \neq h(x), p_1(x) \neq \lambda p_0(x)\}$, $S_3 = \{x : h^*(x) \neq h(x), p_1(x) = \lambda p_0(x)\}$. By Neyman-Pearson Lemma part (ii), we know that $\nu(S_2) = 0$. Moreover, we have

$$\alpha = \int h^* p_0 d\nu$$
$$= \int_{S_1} h^* p_0 d\nu + \int_{S_2} h^* p_0 d\nu + \int_{S_3} h^* p_0 d\nu$$

Since $\nu(S_2) = 0$, we conclude that $\int_{S_2} h^* p_0 d\nu = 0$. Furthermore, on $S_3$ we have $h \equiv 1$ and $h \neq h^*$. So $h^* \equiv 0$ on $S_3$ and $\int_{S_3} h^* p_0 d\nu = 0$. Hence,

$$\alpha = \int_{S_1} h^* p_0 d\nu = \int_{S_1} h p_0 d\nu$$

On the other hand, we have

$$\alpha = \int h p_0 d\nu = \int_{S_1} h p_0 d\nu + \int_{S_2} h p_0 d\nu + \int_{S_3} h p_0 d\nu$$
$$= \alpha + \int_{S_3} h p_0 d\nu$$

Therefore, $\int_{S_3} h p_0 d\nu = \int_{S_3} p_0 d\nu = 0$, because $h \equiv 1$ on $S_3$. We claim that $\nu(S_3) = 0$. By contradiction assume that $\nu(S_3) > 0$. First we show that $p_0$ is positive on $S_3$ a.s. $\nu$. The reason is that on $S_3$ we have $h \equiv 1$ and $h \neq h^*$. In addition, for $x$ satisfying $p_0(x) = 0$, we have $h(x) = h^*(x) = 1$ a.s. $\nu$. Therefore, $p_0$ must be positive on $S_3$ a.s. $\nu$. However, we have $\int_{S_3} p_0 d\nu = 0$ which cannot be true since $\nu(S_3) > 0$ and $p_0 > 0$ a.s. $\nu$ on $S_3$. Hence, we conclude that $\nu(S_3) = 0$. Finally, we obtain that $\nu(S_2 \cup S_3) = 0$, where $S_2 \cup S_3 = \{x : h^*(x) \neq h(x)\}$. □

**Remark A.2.** *Proposition A.2 implies that $\mathcal{L}^S(\alpha)$ in Definition 3.5 is unique a.s. $\nu$.*

Now we are ready to prove Proposition 3.6 in Section 3, which characterizes equivalent source and target. First we show the following technical lemma.

**Lemma A.3.** *Suppose that $\alpha$ is achievable. If $\alpha < 1$, then (2.2) cannot have any other solution $h$ with $R_{\mu_0}(h) < \alpha$ and $R_{\mu_{1,S}}(h) = 0$.*

*Proof of Lemma A.3.* By contradiction, suppose that there exists an optimal solution $h_1$ of (2.2) with $R_{\mu_0}(h_1) < \alpha$ and $R_{\mu_{1,S}}(h_1) = 0$. Furthermore, let $h = \mathbb{1}_{\mathcal{L}^S(\alpha)}$. Since both $h$ and $h_1$ are optimal solutions of (2.2), we have $R_{\mu_{1,S}}(h) = R_{\mu_{1,S}}(h_1) = 0$. By Neyman-Pearson Lemma part (ii), we know that $h_1 = h$ on $\{x : p_0(x) \neq \lambda p_{1,S}(x)\}$ a.s. $\nu$. Let us define the sets $S_1 = \{x : h_1(x) \neq h(x)\}$ and $S_2 = \{x : p_0(x) = \lambda p_{1,S}(x)\}$. Then we have $S_1 \subset S_2$ a.s. $\nu$. Since $h \equiv 1$ on $S_2$, we must have $h_1 \equiv 0$ on $S_1$ a.s. $\nu$. From $R_{\mu_{1,S}}(h) = R_{\mu_{1,S}}(h_1) = 0$ we conclude that $\mu_{1,S}(S_1) = 0$ and from $R_{\mu_0}(h_1) < R_{\mu_0}(h) = \alpha$ we conclude that $\mu_0(S_1) > 0$. Let $S_1' = \{x \in S_1 : p_0(x) > 0\}$. Hence $\mu_0(S_1') > 0$ and $\nu(S_1') > 0$. Then, let us define the set $A = \{x : x \in S_1', p_{1,S}(x) = 0\}$. We have

$$\mu_{1,S}(S_1') = \int_{S_1'} p_{1,S} d\nu = \int_{S_1' \backslash A} p_{1,S} d\nu = 0.$$

Since $p_{1_S} > 0$ on $S_1 \backslash A$, we conclude that $\nu(S_1' \backslash A) = 0$ which implies that $\nu(A) > 0$. On $A$, $p_{1,S} \equiv 0$ and $p_0 > 0$ and $h \equiv 1$ a.s. $\nu$. Hence, we should have $\lambda = 0$ which implies that $\mu_0(\mathcal{L}^S(\alpha)) = 1$. However, we assumed that $\alpha < 1$. $\qquad\square$

*Proof of Proposition 3.6.* (Sufficiency) Suppose that $\mathcal{L}^S(\alpha) \in \{\mathcal{L}_\lambda^T\}_{\lambda \geq 0}$ a.s. $\nu$. Due to Neyman-Pearson Lemma part (i), $\mathbb{1}_{\mathcal{L}^S(\alpha)}$ is an optimal solution of (2.2). Let $h^S \in \mathcal{U}^*$ be any arbitrary optimal solution of (2.2). First we consider the case that $R_{\mu_{1,S}}(h^S) > 0$ (or the power of $h^S$ in the source problem is less than 1). By Neyman-Pearson Lemma part (ii) we have $R_{\mu_0}(h^S) = \alpha$. Then by Proposition A.2, $h^S = \mathbb{1}_{\mathcal{L}^S(\alpha)}$ a.s. $\nu$. We claim that $h^S$ is an optimal solution of (2.3). Since $\mathcal{L}^S(\alpha) \in \{\mathcal{L}_\lambda^T\}_{\lambda \geq 0}$, there exists $\mathcal{L}_{\lambda'}^T$ such that $\mathcal{L}^S(\alpha) = \mathcal{L}_{\lambda'}^T$ a.s. $\nu$. Hence, $\mu_0(\mathcal{L}_{\lambda'}^T) = \alpha$ and $\mathbb{1}_{\mathcal{L}_{\lambda'}^T}$ is an optimal solution of (2.3). Furthermore, $h^S = \mathbb{1}_{\mathcal{L}^S(\alpha)} = \mathbb{1}_{\mathcal{L}_{\lambda'}^T}$ a.s. $\nu$ which implies that $h^S$ is an optimal solution of (2.3).

If $R_{\mu_{1,S}}(h^S) = 0$ and $R_{\mu_0}(h^S) = \alpha$, it would be similar to the previous case. Furthermore, by Lemma A.3, we cannot have a solution $h^S$ with $R_{\mu_{1,S}}(h^S) = 0$ and $R_{\mu_0}(h^S) < \alpha$.

(Necessity) Suppose that any optimal solution of (2.2) is also an optimal solution of (2.3). Since $\mathbb{1}_{\mathcal{L}^S(\alpha)}$ is an optimal solution of (2.2), we conclude that it is also an optimal solution of (2.3). Since $R_{\mu_0}(\mathbb{1}_{\mathcal{L}^S(\alpha)}) = \alpha$, by Proposition A.2 and $\alpha$ achievability, $\mathbb{1}_{\mathcal{L}^T(\alpha)} = \mathbb{1}_{\mathcal{L}^S(\alpha)}$ a.s. $\nu$. Therefore, $\mathcal{L}^S(\alpha) = \mathcal{L}^T(\alpha)$ a.s. $\nu$ and $\mathcal{L}^S(\alpha) \in \{\mathcal{L}_\lambda^T\}_{\lambda \geq 0}$ a.s. $\nu$.

$\qquad\square$

## A.3 EXAMPLE CORRESPONDING TO FIG 3

**Example A.4.** *Let $\nu$ be the Lebesgue measure, $\alpha = \frac{1}{16}$, and $\mathcal{U}$ be all the measurable 0-1 functions on $\mathbb{R}$. Furthermore, let $\mu_{1,S} \sim Unif[1,2]$, $\mu_{1,T} \sim Unif[\frac{4}{3}, \frac{8}{3}]$, and*

$$p_0(x) = \begin{cases} \frac{x}{4} + \frac{1}{2} & -2 \leq x \leq 0 \\ \frac{-x}{4} + \frac{1}{2} & 0 < x \leq 2 \end{cases}$$

*Then, we have $\mathcal{L}^S(\alpha) = \mathcal{L}^T(\alpha) = (-\infty, -2] \cup [\frac{3}{2}, +\infty)$. Consider the hypothesis $h = \mathbb{1}_{\{x \in [\frac{3}{2}, 2]\}}$ which is a solution in the source but not in the target. However, source is equivalent to target under $\mathcal{U}^*$.*

## B    APPENDIX B (OUTLIER TRANSFER EXPONENT)

### B.1    PROOF OF PROPOSITION 4.2

*Proof.* Let $\mu_0 \sim \mathcal{N}(0,1)$, $\mu_{1,S} \sim \mathcal{U}[0,1]$, $\mu_{1,T} \sim A_1 \cdot \mathcal{U}[t_1, 2t_0 - 1] + A_2 \cdot \mathcal{U}[2t_0 - 1, 1]$ where $A_2 > A_1 > 0$, $t_1 > 0$, $\frac{1}{2} < t_0 < 1$, $2t_0 - 1 > t_1$ and $A_1(2t_0 - t_1 - 1) + A_2(2 - 2t_0) = 1$. Moreover, $\mathcal{H} = \{\mathbb{1}_{\{x \in [a,1] \cup [b,t_0]\}}(x) : t_0 \le a \le 1, t_1 \le b \le t_0\}$. Let $\alpha = \mu_0([t_0,1])$ and $r < \mu_0([2t_0 - 1, t_0]) - \alpha$. Clearly by Neyman-Pearson Lemma the unique source and target optimal solutions are $h^*_{S,\alpha} = h^*_{T,\alpha} = \mathbb{1}_{\{t_0 \le x \le 1\}}$. Then for any $h$ with $R_{\mu_0}(h) \le \alpha + r$, $h$ is of the form $h = \mathbb{1}_{\{x \in [a,1] \cup [b,t_0]\}}$ for some $a \in [t_0, 1]$ and $b \in [2t_0 - 1, t_0]$. Hence, $R_{\mu_{1,S}}(h) - R_{\mu_{1,S}}(h^*_{S,\alpha}) = a + b - 2t_0$ and $R_{\mu_{1,T}}(h) - R_{\mu_{1,T}}(h^*_{S,\alpha}) = A_2(a + b - 2t_0)$ which implies that $\rho(r) = 1$. However, if we take $h = \mathbb{1}_{\{2t_0 - 1 - \epsilon \le x \le t_0\}}(x)$ for small enough $0 < \epsilon < \frac{(1-t_0)(A_2 - A_1)}{A_1}$, which violates the condition $R_{\mu_0}(h) \le \alpha + r$, (4.2) does not hold for any $\rho < \infty$.  $\square$

## C    APPENDIX C: PROOF OF THEOREM 4.5 (LOWER BOUND)

### C.1    MINIMAX LOWER BOUND IS LARGER THAN $\min\{(\frac{d_{\mathcal{H}}}{n_S})^{\frac{1}{2\rho}}, (\frac{d_{\mathcal{H}}}{n_T})^{\frac{1}{2}}\}$ FOR $d_{\mathcal{H}} \ge 17$

Let $d = d_{\mathcal{H}} - 1$ and $d_{\mathcal{H}}$ be odd (If $d_{\mathcal{H}}$ is even then define $d = d_{\mathcal{H}} - 2$). Then pick $d_{\mathcal{H}}$ points $\mathcal{S} = \{x_0, x_{1,1}, ..., x_{1, \frac{d}{2}}, x_{2,1}, ..., x_{2, \frac{d}{2}}\}$ from $\mathcal{X}$ shattered by $\mathcal{H}$ (if $d_{\mathcal{H}}$ is even then we pick $d_{\mathcal{H}} - 1$ points). Moreover, let $\tilde{\mathcal{H}}$ be the projection of $\mathcal{H}$ onto the set $\mathcal{S}$ with the constraint that all $h \in \tilde{\mathcal{H}}$ classify $x_0$ as $0$.

Next we construct a distribution $\mu_0$ and a family of pairs of distributions $(\mu^\sigma_{1,S}, \mu^\sigma_{1,T})$ indexed by $\sigma \in \{-1, +1\}^{\frac{d}{2}}$. In the following, we fix $\epsilon = c_1 \cdot \min\{(\frac{d_{\mathcal{H}}}{n_S})^{\frac{1}{2\rho}}, (\frac{d_{\mathcal{H}}}{n_T})^{\frac{1}{2}}\}$ for a constant $c_1 < 1$ to be determined.

**Distribution $\mu_0$:** We define $\mu_0$ on $\mathcal{S}$ as follows:

$$\mu_0(x_{1,i}) = \mu_0(x_{2,i}) = \frac{2\alpha}{d} \quad \text{for } i = 1, ..., \frac{d}{2}$$

and $\mu_0(x_0) = 1 - 2\alpha$.

**Distribution $\mu^\sigma_{1,T}$:** We define $\mu^\sigma_{1,T}$ on $\mathcal{S}$ as follows:

$$\mu^\sigma_{1,T}(x_{1,i}) = \frac{1}{d} + (\sigma_i/2) \cdot \frac{\epsilon}{d} \quad \text{for } i = 1, ..., \frac{d}{2}$$
$$\mu^\sigma_{1,T}(x_{2,i}) = \frac{1}{d} - (\sigma_i/2) \cdot \frac{\epsilon}{d} \quad \text{for } i = 1, ..., \frac{d}{2}$$

and $\mu^\sigma_{1_T}(x_0) = 0$.

**Distribution $\mu^\sigma_{1,S}$:** We define $\mu^\sigma_{1,S}$ on $\mathcal{S}$ as follows:

$$\mu^\sigma_{1,S}(x_{1,i}) = \frac{1}{d} + (\sigma_i/2) \cdot \frac{\epsilon^\rho}{d} \quad \text{for } i = 1, ..., \frac{d}{2}$$
$$\mu^\sigma_{1,S}(x_{2,i}) = \frac{1}{d} - (\sigma_i/2) \cdot \frac{\epsilon^\rho}{d} \quad \text{for } i = 1, ..., \frac{d}{2}$$

and $\mu^\sigma_{1_S}(x_0) = 0$.

**Verifying the transfer distance condition.** For any $\sigma \in \{-1, +1\}^{\frac{d}{2}}$, let $h_\sigma \in \tilde{\mathcal{H}}$ be the minimizer of $R_{\mu^\sigma_{1,S}}$ and $R_{\mu^\sigma_{1,T}}$ with type-I error w.r.t. $\mu_0$ at most $\alpha$. Then $h_\sigma$ satisfies the following:

$$h_\sigma(x_{1,i}) = 1 - h_\sigma(x_{2,i}) = \begin{cases} 1 & \text{if } \sigma_i = 1 \\ 0 & \text{otherwise} \end{cases} \quad \text{for } i = 1, ..., \frac{d}{2}$$

For any $\hat{h} \in \tilde{\mathcal{H}}$ with $\alpha - \frac{2\alpha}{d} < \mu_0(\hat{h}) < \alpha + \frac{2\alpha}{d}$, we have

$$\mu_{1,T}(h_\sigma = 1) - \mu_{1,T}(\hat{h} = 1) = \frac{k}{d} \cdot \epsilon$$

$$\mu_{1,S}(h_\sigma = 1) - \mu_{1,S}(\hat{h} = 1) = \frac{k}{d} \cdot \epsilon^\rho$$

for some non-negative integer $k \le \frac{d}{2}$. So the outlier transfer exponent is $\rho$ with $C_\rho = 1$. The condition is also satisfied for $\hat{h} \in \tilde{\mathcal{H}}$ with $\mu_0(\hat{h}) \le \alpha - \frac{2\alpha}{d}$. In this case we have

$$\mu_{1_T}(h_\sigma = 1) - \mu_{1_T}(\hat{h} = 1) = \frac{k_1}{d} + \frac{k_2 \epsilon}{2d}$$

$$\mu_{1_S}(h_\sigma = 1) - \mu_{1_S}(\hat{h} = 1) = \frac{k_1}{d} + \frac{k_2 \epsilon^\rho}{2d}$$

for some integers $k_1 \le \frac{d}{2}$ and $k_2 \le d$. Using inequality $(a+b)^\rho \le 2^{\rho-1}(a^\rho + b^\rho)$ the condition can be easily verified.

**Reduction to a packing.** Any classifier $\hat{h} : \mathcal{S} \to \{0,1\}$ can be reduced to a binary sequence in the domain $\{-1,+1\}^d$. We can first map $\hat{h}$ to $(\hat{h}(x_{1,1}), \hat{h}(x_{1,2}), ..., \hat{h}(x_{1,\frac{d}{2}}), \hat{h}(x_{2,1}), ..., \hat{h}(x_{2,\frac{d}{2}}))$ and then convert any element $0$ to $-1$. We choose the Hamming distance as the distance required in Theorem 5.1. By applying Proposition 5.2 we can get a subset $\Sigma$ of $\{-1,+1\}^{\frac{d}{2}}$ with $|\Sigma| = M \ge 2^{d/16}$ such that the hamming distance of any two $\sigma, \sigma' \in \Sigma$ is at least $d/16$. Any $\sigma, \sigma' \in \Sigma$ can be mapped to binary sequences in the domain $\{+1,-1\}^d$ by replicating and negating, i.e., $(\sigma, -\sigma), (\sigma', -\sigma') \in \{+1,-1\}^d$ and the hamming distance of resulting sequences in the domain $\{+1,-1\}^d$ is at least $d/8$. Then for any $\hat{h} \in \tilde{\mathcal{H}}$ with $\mu_0(\hat{h} = 1) < \alpha + \frac{2\alpha}{d}$ and $\sigma \in \Sigma$, if the hamming distance of the corresponding binary sequence of $\hat{h}$ and $\sigma$ in the domain $\{+1,-1\}^d$ is at least $d/8$ then we have

$$\mu_{1,T}(h_\sigma = 1) - \mu_{1,T}(\hat{h} = 1) \ge \frac{d}{8} \cdot \frac{\epsilon}{d} = \frac{\epsilon}{8}$$

In particular, for any $\sigma, \sigma' \in \Sigma$ we have

$$\mu_{1,T}(h_\sigma = 1) - \mu_{1,T}(h_{\sigma'} = 1) \ge \frac{d}{8} \cdot \frac{\epsilon}{d} = \frac{\epsilon}{8}$$

**KL divergence bound.** Define $\Pi_\sigma = (\mu_{1,S}^\sigma)^{n_S} \times (\mu_{1,T}^\sigma)^{n_T}$. For any $\sigma, \sigma' \in \Sigma$, our aim is to bound the KL divergence of $\Pi_\sigma, \Pi_{\sigma'}$. We have

$$\mathcal{D}_{kl}(\Pi_\sigma | \Pi_{\sigma'}) = n_S \cdot \mathcal{D}_{kl}(\mu_{1,S}^\sigma | \mu_{1,S}^{\sigma'}) + n_T \cdot \mathcal{D}_{kl}(\mu_{1,T}^\sigma | \mu_{1,T}^{\sigma'})$$

The distribution $\mu_{1,S}^\sigma$ can be expressed as $P_X^\sigma \times P_{Y|X}^\sigma$ where $P_X^\sigma$ is a uniform distribution over the set $\{1, 2, ..., \frac{d}{2}\}$ and $P_{Y|X=i}^\sigma$ is a Bernoulli distribution with parameter $\frac{1}{2} + \frac{1}{2} \cdot (\sigma_i/2) \cdot \epsilon^\rho$. Hence we get

$$\mathcal{D}_{kl}(\mu_{1,S}^\sigma | \mu_{1,S}^{\sigma'}) = \sum_{i=1}^{\frac{d}{2}} \frac{1}{d/2} \cdot \mathcal{D}_{kl}\left( \text{Ber}(\frac{1}{2} + \frac{1}{2} \cdot (\sigma_i/2) \cdot \epsilon^\rho) | \text{Ber}(\frac{1}{2} + \frac{1}{2} \cdot (\sigma_i'/2) \cdot \epsilon^\rho) \right)$$

$$\le c_0 \cdot \frac{1}{4} \cdot \epsilon^{2\rho}$$

$$\le \frac{1}{4} c_0 c_1^{2\rho} \cdot \frac{d_{\mathcal{H}}}{n_S}$$

$$\le c_0 c_1^{2\rho} \cdot \frac{d}{n_S} \tag{C.1}$$

for some numerical constant $c_0$. Using the same argument we can obtain $\mathcal{D}_{kl}(\mu_{1,T}^\sigma | \mu_{1,T}^{\sigma'}) \le c_0 c_1^2 \cdot \frac{d}{n_T}$. Hence we get

$$\mathcal{D}_{kl}(\Pi_\sigma | \Pi_{\sigma'}) \le 2 c_0 c_1 d.$$

Then, for sufficiently small $c_1$ we get $\mathcal{D}_{kl}(\Pi_\sigma | \Pi_{\sigma'}) \le \frac{1}{8} \log M$ which satisfies condition (ii) in Proposition 5.1.

Therefore, for any learner that outputs a hypothesis $\hat{h}$ from $\{h \in \mathcal{H} : \mu_0(h) \le \alpha + \frac{2\alpha}{d_\mathcal{H}}\}$ with probability $1 - \delta_0$, there exist $(\mu_0, \mu_{1,S}, \mu_{1,T}) \in \mathcal{F}_\mathcal{H}(\rho, \alpha, 1, \Delta)$ such that condition on $\hat{h} \in \{h \in \mathcal{H} : \mu_0(h) \le \alpha + \frac{2\alpha}{d_\mathcal{H}}\}$ we have

$$\mathbb{P}_{S_{\mu_0}, S_{\mu_{1,S}}, S_{\mu_{1,T}}} \left( \mathcal{E}_{1,T}(\hat{h}) > c \cdot \min\{\Delta + (\frac{d_\mathcal{H}}{n_S})^{\frac{1}{2\rho}}, (\frac{d_\mathcal{H}}{n_T})^{\frac{1}{2}}\} \right) \ge c'$$

which implies that the unconditional probability is as follows

$$\mathbb{P}_{S_{\mu_0}, S_{\mu_{1,S}}, S_{\mu_{1,T}}} \left( \mathcal{E}_{1,T}(\hat{h}) > c \cdot \min\{\Delta + (\frac{d_\mathcal{H}}{n_S})^{\frac{1}{2\rho}}, (\frac{d_\mathcal{H}}{n_T})^{\frac{1}{2}}\} \right) \ge (1 - \delta_0)c' \ge c''$$

## C.2 MINIMAX LOWER BOUND IS LARGER THAN $\min\{(\frac{d_\mathcal{H}}{n_S})^{\frac{1}{2\rho}}, (\frac{d_\mathcal{H}}{n_T})^{\frac{1}{2}}\}$ FOR $16 \ge d_\mathcal{H} \ge 3$

Pick three points $\mathcal{S} = \{x_0, x_1, x_2\}$ from $\mathcal{X}$ shattered by $\mathcal{H}$. Then we construct a distribution $\mu_0$ and two pairs of distributions $(\mu_{1,S}^k, \mu_{1,T}^k)$ for $k = -1, 1$. Also fix $\epsilon = c_1 \cdot \min\{(\frac{1}{n_S})^{\frac{1}{2\rho}}, (\frac{1}{n_T})^{\frac{1}{2}}\}$ for a constant $c_1 < 1$ to be determined.

**Distribution $\mu_0$:** We define $\mu_0$ on $\mathcal{S}$ as follows:

$$\mu_0(x_0) = 1 - 2\alpha, \quad \mu_0(x_1) = \mu_0(x_2) = \alpha$$

**Distribution $\mu_{1,T}^k$:** We define $\mu_{1,T}^k$ on $\mathcal{S}$ as follows:

$$\mu_{1,T}^k(x_0) = 0, \quad \mu_{1,T}^k(x_1) = \frac{1}{2} + \frac{k}{2} \cdot \epsilon, \quad \mu_{1,T}^k(x_2) = \frac{1}{2} - \frac{k}{2} \cdot \epsilon$$

**Distribution $\mu_{1,S}^k$:** We define $\mu_{1,S}^k$ on $\mathcal{S}$ as follows:

$$\mu_{1,S}^k(x_0) = 0, \quad \mu_{1,S}^k(x_1) = \frac{1}{2} + \frac{k}{2} \cdot \epsilon^\rho, \quad \mu_{1,S}^k(x_2) = \frac{1}{2} - \frac{k}{2} \cdot \epsilon^\rho$$

Let $\Pi_k = (\mu_{1,S}^k)^{n_S} \times (\mu_{1,T}^k)^{n_T}$ for $k = -1, 1$. Then using the same argument we get $\mathcal{D}_{kl}(\Pi_{-1} | \Pi_1) \le c$ where $c$ is a numerical constant. Furthermore, let $h_k$ be the optimal solution with type-I error at most $\alpha$ for the distributions $(\mu_0, \mu_{1,S}^k)$ and $(\mu_0, \mu_{1,T}^k)$. It is easy to see that $R_{\mu_{1,T}^k}(h_{-k}) - R_{\mu_{1,T}^k}(h_k) = \epsilon$.

Using Le Cam's method we get that for any $\hat{h}$ chosen from $\mathcal{H}_\alpha = \{h \in \mathcal{H} : \mu_0(h) \le \alpha + \frac{2\alpha}{3}\}$ there exist $(\mu_0, \mu_{1,S}, \mu_{1,T}) \in \mathcal{F}_\mathcal{H}(\rho, \alpha, 1, 0)$ such that

$$\mathbb{P}_{S_{\mu_0}, S_{\mu_{1,S}}, S_{\mu_{1,T}}} \left( \mathcal{E}_{1,T}(\hat{h}) > c \cdot \min\{(\frac{1}{n_S})^{\frac{1}{2\rho}}, (\frac{1}{n_T})^{\frac{1}{2}}\} \right) \ge c'$$

Since $d_\mathcal{H} \le 16$ we conclude that

$$\mathbb{P}_{S_{\mu_0}, S_{\mu_{1,S}}, S_{\mu_{1,T}}} \left( \mathcal{E}_{1,T}(\hat{h}) > c \cdot \min\{(\frac{d_\mathcal{H}}{n_S})^{\frac{1}{2\rho}}, (\frac{d_\mathcal{H}}{n_T})^{\frac{1}{2}}\} \right) \ge c'$$

for some numerical constants $c, c'$.

## C.3 MINIMAX LOWER BOUND IS LARGER THAN $\min\{\Delta, (\frac{d_\mathcal{H}}{n_T})^{\frac{1}{2}}\}$

We only show it for the case where $d_\mathcal{H} \ge 17$. The other case follows the same idea as in Section C.2.

We follow the same idea as in the previous part. Let $\epsilon = c_1 \cdot \min\{\Delta, (\frac{d_\mathcal{H}}{n_T})^{\frac{1}{2}}\}$ and pick the same set $\mathcal{S}$ from $\mathcal{X}$ shattered by $\mathcal{H}$ construct the distributions on $\mathcal{S}$ as follows:

**Distribution $\mu_0$:** We define $\mu_0$ on $\mathcal{S}$ as follows:

$$\mu_0(x_{1,i}) = \mu_0(x_{2,i}) = \frac{2\alpha}{d} \quad \text{for} \quad i = 1, ..., \frac{d}{2}$$

and $\mu_0(x_0) = 1 - 2\alpha$.

**Distribution $\mu_{1,T}^\sigma$:** We define $\mu_{1,T}^\sigma$ on $\mathcal{S}$ as follows:

$$\mu_{1,T}^\sigma(x_{1,i}) = \frac{1}{d} + (\sigma_i/2) \cdot \frac{\epsilon}{d} \quad \text{for} \quad i = 1, ..., \frac{d}{2}$$

$$\mu_{1,T}^\sigma(x_{2,i}) = \frac{1}{d} - (\sigma_i/2) \cdot \frac{\epsilon}{d} \quad \text{for} \quad i = 1, ..., \frac{d}{2}$$

and $\mu_{1,T}^\sigma(x_0) = 0$.

**Distribution $\mu_{1,S}^\sigma$:** We define $\mu_{1,S}^\sigma$ on $\mathcal{S}$ as follows:

$$\mu_{1,S}^\sigma(x_{1,i}) = \frac{1}{d} + (1/2) \cdot \frac{\epsilon^\rho}{d} \quad \text{for} \quad i = 1, ..., \frac{d}{2}$$

$$\mu_{1,S}^\sigma(x_{2,i}) = \frac{1}{d} - (1/2) \cdot \frac{\epsilon^\rho}{d} \quad \text{for} \quad i = 1, ..., \frac{d}{2}$$

and $\mu_{1,S}^\sigma(x_0) = 0$.

Note that unlike previous part, all the distributions $\mu_{1_S}^\sigma$ are the same for different $\sigma$'s.

**Verifying $\mathcal{E}_{1,T}(h_{S,\alpha}^*) \le \Delta$.** For every pair of $(\mu_{1,S}^\sigma, \mu_{1,T}^\sigma)$ we have

$$\mathcal{E}_{1,T}(h_{S,\alpha}^*) \le \frac{d}{2} \cdot \frac{\epsilon}{d} \le \Delta$$

verifying the transfer distance condition and reducing to a packing parts follow the same idea. We just bound the corresponding kL-divergence.

**KL divergence bound.** Define $\Pi_\sigma = (\mu_{1,S}^\sigma)^{n_S} \times (\mu_{1,T}^\sigma)^{n_T}$. We have

$$\mathcal{D}_{kl}(\Pi_\sigma | \Pi_{\sigma'}) = n_S \cdot \mathcal{D}_{kl}(\mu_{1,S}^\sigma | \mu_{1_S}^{\sigma'}) + n_T \cdot \mathcal{D}_{kl}(\mu_{1,T}^\sigma | \mu_{1,T}^{\sigma'})$$

Since source distributions are the same, the first term is zero. Following the same argument we get

$$\mathcal{D}_{kl}(\mu_{1,T}^\sigma | \mu_{1,T}^{\sigma'}) \le c_0 \epsilon^2 \le c_0 c_1 \frac{d}{n_T}$$

where $c_0$ is the same numerical constant used in (C.1). Then for sufficiently small $c_1$ we get $\mathcal{D}_{kl}(\Pi_\sigma | \Pi_{\sigma'}) \le \frac{1}{8} \log M$ which satisfies condition (ii) in Proposition 5.1.

