# OpenReview forum: "Tight Rates in Supervised Outlier Transfer Learning"
_ICLR.cc/2024/Conference — ICLR 2024 poster_

### Official Review · Reviewer_kTQ3 · 2023-10-31

**Soundness:** 2 fair
**Presentation:** 2 fair
**Contribution:** 2 fair
**Rating:** 5
**Confidence:** 3

**Summary:**

This paper studies the outlier transfer problem, that is, the problem of transfer learning under the setting of outlier detection or rare class classification. The objective of the Neyman-Pearson classification problem, which formalizes the rare class classification problem, is to achieve low classification error on the rare class, while keeping the classification error on the common class under a threshold. However, in practice, we usually have only limited amount of or even none data from the target rare class, but some data from a related source rare class. This is where transfer learning comes into play. The goal of this paper is to theoretically understand when and how the knowledge from a source class can improve the classification performance on a target class under the setting of rare class classification.

The authors first show that at the population level, under certain assumptions, all the solutions to the source Neyman-Pearson classification problem are also solutions to the target Neyman-Pearson classification problem. Then the authors turn their attention to the finite-sample setting. The authors first define the outlier transfer exponent, which is a notion of discrepancy between source and target under a hypothesis class. With that discrepancy, the authors give a minimax lower bound on the target-excess error, which measures the difference between the expected error of the solution obtained by transfer learning and of the optimal solution. Furthermore, the authors propose an algorithm that does not need any prior knowledge of the discrepancy between the source and target class distributions.

**Strengths:**

1. The paper studied an important practical problem.

**Weaknesses:**

(1) A lower bound on the target-excess error is not as informative as an upper bound. Is it possible to derive an upper bound on the target-excess error under appropriate conditions?

(2) The algorithm proposed in Section 4.8 requires as input the VC dimension of the hypothesis class. However, in practice, the exact VC dimension may be unknown. Could you please give some practical suggestions on using this algorithm when the exact VC dimension is unknown?

(3) The notation in inequality (4.1) is a little redundant. Since $h_{S, \alpha}^*$ is a solution to the source problem, the difference between the expected error of any $h$ in the hypothesis class and of $h_{S, \alpha}^*$ w.r.t. the source distribution must be non-negative. So, there is no need to use the max function.

 (4) The theoretical results mainly rely on the previous techniques.

(5) There is no experiment.

**Questions:**

There are several typos, including:

(1) Page 4, in the 5th line in Section 3, the source and target problem are denoted by the same notation.

(2) Page 4, in the last line, the LHS and RHS of the second to last inequality are the same.

(3) Page 6, in the 5th line in Section 4.3, $n_S$ should be $n_T$.

---

> ### Author Response · Authors · 2023-11-16
>
> 1. A lower bound on the target-excess error is not as informative as an upper bound. Is it possible to derive an upper bound on the target-excess error under appropriate conditions?
>
> It seems that there might have been a misunderstanding on the reviewer's side: we actually provide matching upper-bound (Theorem 4.6 in Section 4.8), and even more so, show that the bound can be achieved adaptively, i.e., with no prior knowledge of the discrepancy between distributions. Together with the lower-bound, our results therefore establish information-theoretic limits for the problem. We emphasize that the lower-bound in fact holds very generally, against any hypothesis class (of VC $\geq$ 3), and any value of the discrepancy (transfer-exponent) and therefore tightly captures the problem.
>
> 2. The algorithm requires knowledge of the VC dimension.
>
> We first note that the assumption of known VC is common in theoretical works and in fact we know very few results where such an assumption is not made. This is because, the exact VC or an upper-bound thereof is known for many of the common hypothesis classes encountered in the theoretical literature, e.g., linear to polynomial classes, classification trees, fixed architecture Neural networks, etc.
>
> However, we agree with the reviewer that in practice, we might not know the tightest upper-bound for a given choice of models, and theory works on uniform concentration then prescribes the use of notions such as Empirical Rademacher Complexity (ERC) which may be estimable in principle (but often not practical). All our upper bounds hold also under ERC (this is evident in our concentration arguments), and we will make sure to emphasize this.
>
> 3. The notation in inequality (4.1) is a little redundant. Since $h^*_{S,\alpha}$ is a solution to the source problem... there is no need to use the max function.
>
> It appears that the reviewer must have missed a nuance in the definition; the notation is not actually redundant: $h^*_{S,\alpha}$ is the best classifier among $\mathcal{H}_{\alpha} = \\{h\in \mathcal{H}: R_0(h)\leq \alpha\\}\$.
>
> However, in the definition, $h$ belongs to a larger class, namely $ \mathcal{H}_{\alpha+r} =\\{h\in \mathcal{H}: R_0(h)\leq \alpha+r\\}$. Hence, the difference of the risks can be negative.
>
> 4. The theoretical results mainly rely on the previous techniques.
>
> Neyman-Pearson classification, as discussed in the first part of the paper, is inherently different from traditional classification. Already, as argued in the first set of results, widely differing distributions may admit the same optimal classifiers (see Section 3).
> More generally, while some of the tools may appear similar, the analysis requires fundamentally different approaches. For instance a main difficulty arises from the fact that excess risks could be negative as explained above, due to the fact that a learner may operate on a different subset $\mathcal{H}_{a+r}$ of $\mathcal{H}$ since it can only estimate $\mathcal{H}_a$ from data.
>
> 5. There is no experiment.
>
> We refer the reviewer to our answer to a similar question by Reviewer 3fJQ (question 2). Namely, we do not claim to propose a new procedure. Rather the procedure is of a theoretical nature and mainly serves to establish that the lower-bound of Theorem 4.4 is attainable in principle, even without prior knowledge of distributional conditions. We re-emphasize that our main aim in this work is theoretical, and is to shed some needed insights on the limits of performance in outlier transfer.
>
>
> Questions:
>
> There are several typos:
>
> Thanks for bringing the typos to our attention. We fixed them and uploaded a revised version.

---

> > ### Comment · Reviewer_kTQ3 · 2023-11-23
> > **Thanks**
> >
> > Thanks for your response, and I will consider it in the next discussion stage.

---

### Official Review · Reviewer_3fJQ · 2023-11-01

**Soundness:** 4 excellent
**Presentation:** 4 excellent
**Contribution:** 2 fair
**Rating:** 6
**Confidence:** 4

**Summary:**

This paper provides a rigorous theoretical analysis of transfer learning in outlier detection. It first considers a simplified setting in which the optimal outlier classifier is the same between source and target distributions to illustrate how outlier detection differs from standard classification. The paper then addresses the much more difficult setting in which the outlier classifiers could differ, proposing an adaptive algorithm with a theoretical guarantee.

**Strengths:**

- The paper is very well-presented.
- The theory is compelling and elegant.
- I think that the "same optimal classifier" setting between source and target distributions seems unrealistic (e.g., the setting of Figure 1) but I can see why from a theoretical standpoint, analyzing this simpler setting is a good starting point and already there are interesting insights, especially in contrasting this outlier setup to traditional classification.
- The extension of the transfer exponent to the outlier setting is a valuable contribution.

**Weaknesses:**

- As far as I can tell, this paper does not actually follow the ICLR LaTeX template. For instance, the margins don't appear correct? Please fix this.
- There are no numerical experiments. I think this paper would improve dramatically with experimental results, especially on real data, and especially on showing how well the adaptive method in Section 4.8 works in practice.
- Detailed discussion of how applied researchers address this outlier transfer problem in practice would be helpful to provide some point of reference (even if these existing approaches lack guarantees): for instance, even getting a rough understanding of whether there are common conceptual ideas used would be helpful or if actually the methods are just completely different (if so, maybe some discussion of what the key conceptual differences are would be helpful).

**Questions:**

See "weaknesses".

---

> ### Author Response · Authors · 2023-11-16
>
> 1. This paper does not actually follow the ICLR LaTeX template. For instance, the margins don't appear correct?
>
> Thank you for pointing this out; we had not noticed at first that one of the packages we had included unfortunately interacted with the ICLR margin. We have now corrected that and uploaded a revised version.
>
> 2. I think this paper would improve dramatically with experimental results ... especially on showing how well the adaptive method in Section 4.8 works in practice.
>
> The adaptive method is in fact of a theoretical nature and not practical at the moment since it relies on 0-1 risk minimization. It only serves to support our theoretical findings, namely that, even without knowledge of the discrepancy between distributions, it is possible in principle to achieve nearly the same optimal rate as an oracle. As a first theoretical work on the subject of outlier transfer, our aim is to yield such initial insights towards developing practical procedures with similar guarantees, e.g., by relying on properly designed surrogate losses for N-P classification. We do hope however that the reviewer will still appreciate the solid foundation we are laying towards such a practical goal. Our main aims at this point are theoretical---i.e., understanding achievable rates in this under-studied transfer setting---and we therefore never claimed to be proposing an algorithm at this point. We will do our best to clarify this further.
>
> 3. Detailed discussion of how applied researchers address this outlier transfer problem in practice would be helpful ...
>
> Various heuristics have been proposed so far, most of which boiling down to the simplest idea of combining both samples and hoping for the best; as such very little is understood at the moment, which motivates our work. We agree with you that such discussion will really enhance the paper, and we will follow your suggestion.

---

### Official Review · Reviewer_Ag52 · 2023-11-01

**Soundness:** 3 good
**Presentation:** 4 excellent
**Contribution:** 4 excellent
**Rating:** 8
**Confidence:** 4

**Summary:**

The paper adopts the traditional framework of Neyman-Pearson classification to formalize supervised outlier detection of transfer learning. The added assumption is that one has access to some related but
imperfect outlier data. The authors first determine the information-theoretic limits of the problem. Next, they also show that, in principle, these information-theoretic limits are achievable by adaptive procedures.

**Strengths:**

1. The outlier detection in transfer learning is an interesting and valuable topic in the learning community.

2. The literature part is very clear.

3. The structure of the paper is easy to follow.

4. The setup of the paper is clear

5. The paper provided solid theoretic results on the minimax bounds and rates.

**Weaknesses:**

1. Only finite-sample results are provided. There is no further analysis of asymptotic properties on the large dataset.

**Questions:**

1. If the size is large, will the results have special asymptotic properties?

---

> ### Author Response · Authors · 2023-11-16
>
> 1. Only finite-sample results are provided... will the results have special asymptotic properties?
>
>  Thank you for the very nuanced question. Unfortunately, besides the fact that our finite-sample results imply consistency, in fact, strong consistency (since success probabilities are exponential), we did not consider other notions of asymptotic convergence, so it is unclear to us at this point whether any special asymptotic phenomena may arise.

---

> > ### Comment · Reviewer_Ag52 · 2023-11-23
> >
> > Thank you for your response. I'm confused about the claim of "strong consistency (since success probabilities are exponential)" in the main results in Theorem 4.4 and 4.6. As no asymptotic property (especially convergence rate) is provided, it would limit the theoretical contribution.
> >
> > In addition, I have also read other reviewers' comments and the author's rebuttal. The first comment from the Reviewer kTQ3 and reply from the authors discussed the upper and lower bounds. I found that the upper bound is not tight. There is a large gap between the lower and upper bound. This gap might also limit the theoretical contribution.
> >
> > In all, I lowered the confidence score to 4.

---

### Meta-Review · Area_Chair_MDe1 · 2023-12-08

**Metareview:**

The authors leverage Neyman-Pearson to study transfer learning in outlier detection tasks. They devise lower and upper bounds and provide formal proofs for their theoretical claims.

**Justification For Why Not Higher Score:**

This is a purely theoretical paper and the problems/misunderstandings at the end of the discussion phase perhaps indicate that the venue is not perfect for the paper. None of the reviewers is very confident about her/his decision which may point to the very same issue.

**Justification For Why Not Lower Score:**

The paper itself appears solid and shouldn't be rejected because of 'missing experiments' or a misunderstanding at the end of the discussion phase. My conclusion is, however, that the target audience of the paper is rather small (otherwise there would have been higher confidences).

---

### Decision · Program_Chairs · 2024-01-16

Accept (poster)